# Graph Auto-Encoder Via Neighborhood Wasserstein Reconstruction

**Mingyue Tang**[1]*,**Carl Yang**[2]*,**Pan Li**[3]†
[1]Department of Engineering Systems and Environment, University of Virginia
[2]Department of Computer Science, Emory University
[3]Department of Computer Science, Purdue University
`utd8hj@virginia.edu, j.carlyang@emory.edu, panli@purdue.edu`

## Abstract

Graph neural networks (GNNs) have drawn significant research attention recently, mostly under the setting of semi-supervised learning. When task-agnostic representations are preferred or supervision is simply unavailable, the auto-encoder framework comes in handy with a natural graph reconstruction objective for unsupervised GNN training. However, existing graph auto-encoders are designed to reconstruct the direct links, so GNNs trained in this way are only optimized towards proximity-oriented graph mining tasks, and will fall short when the topological structures matter. In this work, we revisit the graph encoding process of GNNs which essentially learns to encode the neighborhood information of each node into an embedding vector, and propose a novel graph decoder to reconstruct the entire neighborhood information regarding both proximity and structure via Neighborhood Wasserstein Reconstruction (NWR). Specifically, from the GNN embedding of each node, NWR jointly predicts its node degree and neighbor feature distribution, where the distribution prediction adopts an optimal-transport loss based on the Wasserstein distance. Extensive experiments on both synthetic and real-world network datasets show that the unsupervised node representations learned with NWR have much more advantageous in structure-oriented graph mining tasks, while also achieving competitive performance in proximity-oriented ones.[1]

## 1 Introduction

Network/Graph representation learning (a.k.a. embedding) aims to preserve the high-dimensional complex graph information involving node features and link structures in a low-dimensional embedding space, which requires effective feature selection and dimension reduction (Hamilton et al., 2017b). Graph neural networks (GNNs) have done great jobs to this end, but most of them rely on node labels from specific downstream tasks to be trained in a semi-supervised fashion (Kipf & Welling, 2017; Hamilton et al., 2017a; Wu et al., 2019; Veličković et al., 2018a; Klicpera et al., 2019; Chien et al., 2021). However, similar to other domains, unsupervised representation learning is preferred in many cases, not only because labeled data is not always available (Hu et al., 2020; Xie et al., 2021), but also task-agnostic representations can better transfer and generalize among different scenarios (Erhan et al., 2010; Bengio, 2012; Radford et al., 2016).

To train GNNs in an unsupervised fashion, the classic auto-encoder framework (Baldi, 2012; Goodfellow et al., 2016) provides a natural solution and has been widely explored such as the prominent work (V)GAE (Kipf & Welling, 2016). Specifically, classic auto-encoders aim to decode from the low-dimensional representations information in the entire receptive field of the neural networks. For GNNs, the receptive field of a node representation is its entire neighborhood. However, existing graph auto-encoders appear away from such a motivation and are designed to merely decode the direct links between the node pairs by minimizing a link reconstruction loss. The fundamental difficulty to reconstruct the entire receptive fields of GNNs is due to the non-trivial design of a reconstruction loss on the irregular graph structures. Unfortunately, the over-simplification into link reconstruction makes the learned node representations drop much information and thus provides undesired performance in many downstream tasks.

---

*Equal contribution. †Corresponding author.
[1]Code available at `https://github.com/mtang724/NWR-GAE`.

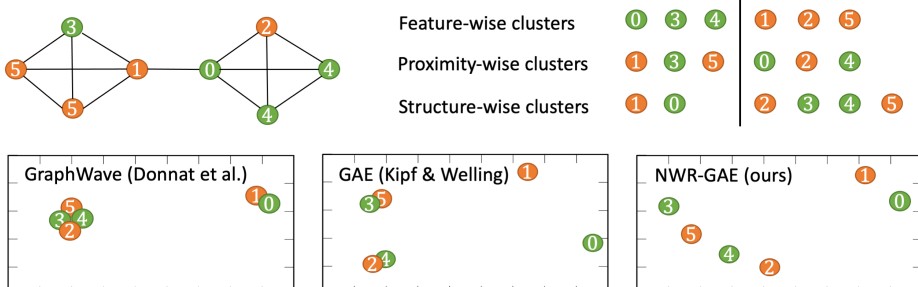

Figure 1: A toy example (redrawn from real experiments) illustrating different types of information in real-world graphs and the two-dimensional node embedding spaces learned by different models. Colors indicates node features. The nodes with same digits are transitive in some graph automorphism.

Take Figure 1 as an example, where different types of information are mixed in a graph (*e.g.*, proximity and structure information as illustrated in Figure 5 in Appendix B (Cui et al., 2021)). The node representations learned by existing graph auto-encoders such as GAE (Kipf & Welling, 2016) are driven too much to be similar on linked nodes due to their simple link reconstruction objective, and thus fail to distinguish node pairs like (2, 4) and (3, 5) in the cliques, though they clearly have different structural roles and node features. On the other hand, structure-oriented embedding models like GraphWave (Donnat et al., 2018) cannot consider node features and spatial proximity, and thus fail to distinguish node pairs like (0, 1), (2, 4) and (3, 5) though they have different features, as well as (2, 5) and (3, 4) though they are further apart. An ideal unsupervised node representation learning model as we advocate in this work is expected to be task-agnostic and encode as much information as possible of all types in a low-dimensional embedding space.

In this work, we aim to fundamentally address the above limitations of existing unsupervised node representation learning models by proposing a novel graph auto-encoder framework for unsupervised GNN training. The new framework is equipped with a powerful decoder that fully reconstructs the information from the entire receptive field of a node representation. Our key technical contribution lies in designing a principled and easy-to-compute loss to reconstruct the entire irregular structures of the node neighborhood. Specifically, we characterize the decoding procedure as iteratively sampling from a series of probability distributions defined over multi-hop neighbors' representations obtained through the GNN encoder. Then, the reconstruction loss can be decomposed into three parts, for sampling numbers (node degrees), neighbor-representation distributions and node features. All of these terms are easy to compute but may represent the entire receptive field of a node instead of just the linkage information to its direct neighbors. For the most novel and important term, neighbor-representation distribution reconstruction, we adopt an optimal-transport loss based on Wasserstein distance (Frogner et al., 2015) and thus name this new framework as Neighborhood Wasserstein Reconstruction Graph Auto-Encoder (`NWR-GAE`). As also illustrated in Figure 1, `NWR-GAE` can effectively distinguish all pairs of nodes dissimilar in different perspectives, and concisely reflect their similarities in the low-dimensional embedding space.

We have conducted extensive experiments on four synthetic datasets and nine real-world datasets. Among the real-world datasets, three have proximity-oriented tasks, three have structure-oriented tasks, and three have proximity-structure-mixed tasks. We can observe significant improvements brought by `NWR-GAE` over the best method among the state-of-the-art baselines on all structure-oriented tasks (8.74% to 18.48%) and proximity-structure-mixed tasks (-2.98% to 8.62%), and competitive performance on proximity-oriented tasks (-3.21% to -0.32%). In-depth ablation and hyper-parameter studies further consolidate the claimed advantages of `NWR-GAE`.

## 2 PRELIMINARIES, MOTIVATIONS & OTHER RELATED WORKS

In this work, we focus on the auto-encoder framework for unsupervised task-agnostic graph representation learning. The original motivation of auto-encoders is to perform neural-network-based dimension reduction of the data that originally lies in a high-dimensional space (Hinton & Salakhutdinov, 2006). Specifically, an auto-encoder consists of two components, an encoder and a decoder. The encoder works to compress each data point into a low-dimensional vector representation, while the decoder works to reconstruct the original information from this vector. By minimizing the reconstruc-

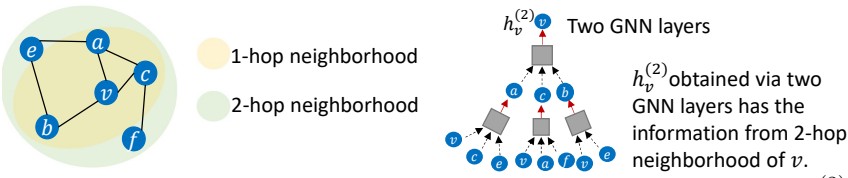

Figure 2: The information source/receptive field of the node representation $h_v^{(2)}$.

tion error, the encoder automatically converges to a good compressor that allows the low-dimensional representations to capture as much information as possible from the original data.

Although the above high-level idea of auto-encoders is clear, when it is applied to graph structured data, the problem becomes challenging. This is because in graph-structured data, information of data points (nodes to be specific as most widely studied) is correlated due to the ambient graph structure. Without a specific task needed in a priori, the learned low-dimensional representation of a node should carry as much information as possible from not only its own features but also the features of the nodes it connects to (both directly and indirectly).

This implies that when building auto-encoders for graph-structure data, we expect the node representations to be able to reconstruct all correlated node features. However, existing graph auto-encoders seem to be away from this motivation. Previous prominent works such as unsupervised Graph-SAGE (Hamilton et al., 2017a), GAE (Kipf & Welling, 2016), their generative variants such as VGAE (Kipf & Welling, 2016), CondGen (Yang et al., 2019) (), and many others (Grover et al., 2019; Pan et al., 2018; Shi et al., 2020; Yang et al., 2021), use GNNs to encode graph structured data into node representations. Without exception, they follow the rationale of traditional network embedding techniques (Perozzi et al., 2014; Qiu et al., 2018; Grover & Leskovec, 2016) and adopt link reconstruction in the decoder as the main drive to optimize their GNN encoders. The obtained node representations best record the network linkage information but lose much of other important information, such as local structures, neighbors' features, etc. Hence, these auto-encoders will most likely fail in other tasks such as node classifications (especially structure-oriented ones as manifested in Figure 1).

To better understand this point, we carefully analyze the source of information encoded in each node representation via a GNN. Suppose a standard message-passing GNN (Gilmer et al., 2017) is adopted as the encoder, which is a general framework that includes GCN (Kipf & Welling, 2017), GraphSAGE (Hamilton et al., 2017a), GAT (Veličković et al., 2018a), GIN (Xu et al., 2019c) and so on. After $k$-hop message passing, the source of information encoded in the representation of a node $v$ essentially comes from the $k$-hop neighborhood of $v$ (Fig. 2). Therefore, a good representation of node $v$ should capture the information of features from all nodes in its $k$-hop neighborhood, which is agnostic to downstream tasks. Note that this may not be ideal as nodes out of $k$-hop neighborhood may also provide useful information, but this is what GNN-based graph auto-encoders can be expected to do due to the architectures of GNN encoders. This observation motivates our study on a novel graph decoder that can better facilitate the goal of GNN-based graph auto-encoders, based on the neighborhood reconstruction principle. We will formalize this principle in Sec. 3.

**Relation to the InfoMax principle.** Recently, DGI (Veličković et al., 2018b), EGI (Zhu et al., 2021) and others (Sun et al., 2020; Hu et al., 2020; You et al., 2020; Hassani & Khasahmadi, 2020; Suresh et al., 2021) have used constrasive learning for unsupervised GNN training methods and may capture information beyond the directed links. They adopt the rule of mutual information maximization (InfoMax), which essentially works to maximize certain correspondence between the learned representations and the original data. For example, DGI (Veličković et al., 2018b) maximizes the correspondence between a node representation and which graph the node belongs to, but this has no guarantee to reconstruct the structural information of node neighborhoods. Recent works even demonstrate that maximizing such correspondence risks capturing only the noisy information that is irrelevant to the downsteam tasks because noisy information itself is sufficient for the models to achieve InfoMax (Tschannen et al., 2020; Suresh et al., 2021), which gets demonstrated again by our experiments. Our goal instead is to let node representations not just capture the information to distinguish nodes but capture as much information as possible to reconstruct the features and structure of the neighborhood.

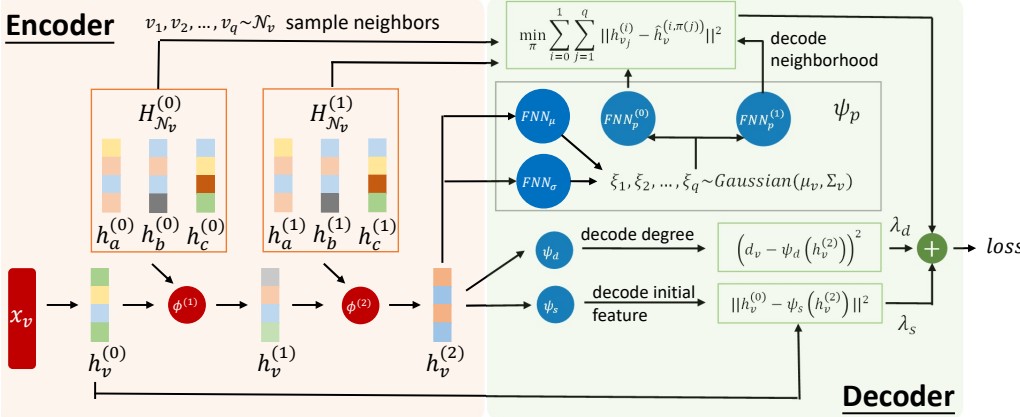

Figure 3: The diagram of our model using an example with 2 GNN layers as the encoder.

**Optimal-transport (OT) losses.** Many machine learning problems depend on the characterization of the distance between two probability measures. The family $f$-divergence has the non-continuous issue when the two measures of interest have non-overlapped support (Ali & Silvey, 1966). Therefore, OT-losses are often adopted and have shown great success in generative models (Gulrajani et al., 2017) and domain adaptation (Courty et al., 2016). OT-losses have been used to build variational auto-encoders for non-graph data (Tolstikhin et al., 2018; Kolouri et al., 2018; Patrini et al., 2020). But one should note the our work is not a graph-data-oriented generalization of these works: They use OT-losses to characterize the distance between the variational distribution and the data empirical distribution while our model even does not use a variational distribution. Our model may be further improved by being reformulated as a variational autoencoder but we leave it as a future direction.

Here, we give a frequently-used OT loss based on 2-Wasserstein distance that will be used later.

**Definition 2.1.** Let $\mathcal{P}, \mathcal{Q}$ denote two probability distributions with finite second moment defined on $\mathcal{Z} \subseteq \mathbb{R}^m$. The 2-Wasserstein distance between $\mathcal{P}$ and $\mathcal{Q}$ defined on $\mathcal{Z}, \mathcal{Z}' \subseteq \mathbb{R}^m$ is the solution to the optimal mass transportation problem with $\ell_2$ transport cost (Villani, 2008):

$$\mathcal{W}_2(\mathcal{P}, \mathcal{Q}) = \left( \inf_{\gamma \in \Gamma(\mathcal{P}, \mathcal{Q})} \int_{\mathcal{Z} \times \mathcal{Z}'} \|Z - Z'\|_2^2 d\gamma(Z, Z') \right)^{1/2} \tag{1}$$

where $\Gamma(\mathcal{P}, \mathcal{Q})$ contains all joint distributions of $(Z, Z')$ with marginals $\mathcal{P}$ and $\mathcal{Q}$ respectively.

## 3 METHODS

### 3.1 NEIGHBORHOOD RECONSTRUCTION PRINCIPLE (NRP)

Let $G = (V, E, X)$ denote the input graph where $V$ is the node set, $E$ is the edge set and $X = \{x_v | v \in V\}$ includes the node features. Given a node $v \in V$, we define its $k$-hop neighborhood as the nodes which have shortest paths of length no greater than $k$ from $v$. Let $\mathcal{N}_v$ denote the 1-hop neighborhood of $v$, and $d_v$ denote the degree of node $v$.

We are to build an auto-encoder to learn a low-dimensional representation of each node. Specifically, the auto-encoder will have an encoder $\phi$ and a decoder $\psi$. $\phi$ could be any message-passing GNNs (Gilmer et al., 2017). The decoder further contains three parts $\psi = (\psi_s, \psi_p, \psi_d)$. The physical meaning of each part will be clear as we introduce them in the later subsections.

**First-hop neighborhood reconstruction.** To simplify the exposition, we start from 1-hop neighborhood reconstruction. We initialize the node representation by $H^{(0)}$ based on $X$. For every node $v \in V$, after being encoded by one GNN layer, its node representation $h_v^{(1)}$ collects information from $h_v^{(0)}$ and its neighbors' representations $H_{\mathcal{N}_v}^{(0)} = \{h_u^{(0)} | u \in \mathcal{N}_v\}$. Hence, we consider the following principle that reconstructs the information from both $h_v^{(0)}$ and $H_{\mathcal{N}_v}^{(0)}$. Therefore, we have

$$\min_{\phi, \psi} \sum_{v \in V} \mathcal{M}((h_v^{(0)}, H_{\mathcal{N}_v}^{(0)}), \psi(h_v^{(1)})), \quad \text{s.t. } h_v^{(1)} = \phi(h_v^{(0)}, H_{\mathcal{N}_v}^{(0)}), \forall v \in V, \tag{2}$$

where $\mathcal{M}(\cdot, \cdot)$ defines the reconstruction loss. $\mathcal{M}$ can be split into two parts that measure self-feature reconstruction and neighborhood reconstruction respectively:

$$\mathcal{M}((h_v^{(0)}, H_{\mathcal{N}_v}^{(0)}), \psi(h_v^{(1)})) = \mathcal{M}_s(h_v^{(0)}, \psi(h_v^{(1)})) + \mathcal{M}_n(H_{\mathcal{N}_v}^{(0)}, \psi(h_v^{(1)})). \tag{3}$$

Note that $\mathcal{M}_s$ works just as the reconstruction loss of a standard feedforward neural network (FNN)-based auto-encoder, so we adopt

$$\mathcal{M}_s(h_v^{(0)}, \psi(h_v^{(1)})) = \|h_v^{(0)} - \psi_s(h_v^{(1)})\|^2. \tag{4}$$

$\mathcal{M}_n$ is much harder to characterize, as it measures the loss to reconstruct a set of features $H_{\mathcal{N}_v}^{(0)}$. There are two fundamental challenges: First, as in real-world networks the distribution of node degrees is often long-tailed, the sets of node neighbors may have very different sizes. Second, to compare even two equal-sized sets, a matching problem has to be solved, as no fixed orders of elements in the sets can be assumed. The complexity to solve a matching problem is high if the size of the set is large. We solve the above challenges by decoupling the neighborhood information into a probability distribution and a sampling number. Specifically, for node $v$ the neighborhood information is represented as an empirical realization of i.i.d. sampling $d_v$ elements from $\mathcal{P}_v^{(0)}$, where $\mathcal{P}_v^{(0)} \triangleq \frac{1}{d_v} \sum_{u \in \mathcal{N}_v} \delta_{h_u^{(0)}}$. Based on this view, we are able to decompose the reconstruction into the part for the node degree and that for the distribution respectively, where the various sizes of node neighborhoods are handled properly. Specifically, we adopt

$$\mathcal{M}_n(H_{\mathcal{N}_v}^{(0)}, \psi(h_v^{(1)})) = (d_v - \psi_d(h_v^{(1)}))^2 + \mathcal{W}_2^2(\mathcal{P}_v^{(0)}, \psi_p^{(1)}(h_v^{(1)})), \tag{5}$$

where $\mathcal{W}_2(\cdot, \cdot)$ is the 2-Wasserstein distance as defined in Eq.1. Plug Eqs.4,5 into Eq.3, we obtain the objective for first-hop neighborhood reconstruction.

**Generalizing to $k$-hop neighborhood reconstruction.** Following the above derivation for the first-hop case, one can similarly generalize the loss for reconstructing $(h_v^{(i-1)}, H_{\mathcal{N}_v}^{(i-1)})$ based on $h_v^{(i)}$ for all $1 \le i \le k$. Then, if we sum such losses over all nodes $v \in V$ and all hops $1 \le i \le k$, we may achieve the objective for $k$-hop neighborhood reconstruction. We may further simplify such an objective. Typically, only the final layer representation $H^{(k)}$ is used as the output dimension-reduced representations. Too many intermediate hops make the model hard to train and slow to converge. Therefore, we adopt a more economic way by merging the multi-step reconstruction:

$$\begin{array}{c} h_v^{(k)} \to h_v^{(k-1)} \to \cdots \to h_v^{(1)} \to h_v^{(0)} \\ \searrow \quad \searrow \qquad \qquad \searrow \\ H_{\mathcal{N}_v}^{(k-1)} \, H_{\mathcal{N}_v}^{(k-2)} \quad \cdots \quad H_{\mathcal{N}_v}^{(0)} \end{array} \quad \implies \quad h_v^{(k)} \xrightarrow[\; H_{\mathcal{N}_v}^{(k-1)} \; H_{\mathcal{N}_v}^{(k-2)} \cdots H_{\mathcal{N}_v}^{(0)} \;]{} h_v^{(0)}$$

That is, we expect $h_v^{(k)}$ to directly reconstruct $H_{\mathcal{N}_v}^{(i)}$ even if $i < k - 1$. Specifically, for each node $v \in V$, we use the reconstruction loss

$$\mathcal{M}'((h_v^{(0)}, \{H_{\mathcal{N}_v}^{(i)} | 0 \le i \le k - 1\}), \psi(h_v^{(k)})) = \mathcal{M}_s(h_v^{(0)}, \psi(h_v^{(k)})) + \sum_{i=0}^{k-1} \mathcal{M}_n(H_{\mathcal{N}_v}^{(i)}, \psi(h_v^{(k)}))$$

$$= \lambda_s \|h_v^{(0)} - \psi_s(h_v^{(k)})\|^2 + \lambda_d (d_v - \psi_d(h_v^{(k)}))^2 + \sum_{i=0}^{k-1} \mathcal{W}_2^2(\mathcal{P}_v^{(i)}, \psi_p^{(i)}(h_v^{(k)})), \tag{6}$$

where $\psi_s$ is to decode the initial features, $\psi_d$ is to decode the degree and $\psi_p^{(i)}, 0 \le i \le k - 1$ is to decode the $i$-layer neighbor representation distribution $\mathcal{P}_v^{(i)}(:\triangleq \frac{1}{d_v} \sum_{u \in \mathcal{N}_v} \delta_{h_u^{(i)}})$. $\lambda_s$ and $\lambda_d$ are non-negative hyperparameters. Hence, the entire objective for $k$-hop neighborhood reconstruction is

$$\min_{\phi, \psi} \sum_{v \in V} \mathcal{M}'((h_v^{(0)}, \{H_{\mathcal{N}_v}^{(i-1)} | 1 \le i \le k\}), \psi(h_v^{(k)})) \quad \text{s.t.} \ H^{(i)} = \phi^{(i)}(H^{(i-1)}), 1 \le i \le k, \tag{7}$$

where $\phi = \{\phi^{(i)} | 1 \le i \le k\}$ include $k$ GNN layers and $\mathcal{M}'$ is defined in Eq.6.

*Remark* 3.1. The loss in Eq. 6 does not directly push the $k$-layer representation $h_v^{(k)}$ to reconstruct the features of all nodes that are direct neighbors but lie in $k$-hop neighborhood of $v$. However, by definition, there exists a shortest path of length no greater than $k$ from $v$ to every node in this $k$-hop neighborhood. As every two adjacent nodes along the path will introduce at least one term of reconstruction loss in the sum of Eq. 6, i.e., the objective in Eq. 7. This guarantees that the optimization in Eq. 7 pushes $h_v^{(k)}$ to reconstruct the entire $k$-hop neighborhood of $v$.

## 3.2 Decoding Distributions — Decoders $\psi_p^{(i)}$, $0 \leq i \leq k-1$

Our NRP essentially represents the node neighborhood $H_{\mathcal{N}_v}^{(i)}$ as a sampling number (node degree $d_v$) plus a distribution of neighbors' representations $\mathcal{P}_v^{(i)}$ (Eqs.5,6). We adopt Wasserstein distance to characterize the distribution reconstruction loss because $\mathcal{P}_v^{(i)}$ has atomic non-zero measure supports in a continuous space, where the family of $f$-divergences such as KL-divergence cannot be applied. Maximum mean discrepancy may be applied but it needs to specify a kernel function.

We set the decoded distribution $\psi_p^{(i)}(h_v^{(k)})$ as an FNN-based transformation of a Gaussian distribution parameterized by $h_v^{(k)}$. The rationale of this setting is due to the universal approximation capability of FNNs to (approximately) reconstruct any distributions in 1-Wasserstein distance, as formally stated in Theorem 3.1 (See the proof in Appendix A, reproduced by Theorem 2.1 in (Lu & Lu, 2020)). For a better empirical performance, our case adopts the 2-Wasserstein distance and an FNN with $m$-dim output instead of the gradient of an FNN with 1-dim outout. Here, the reparameterization trick needs to be used (Kingma & Welling, 2014), i.e.,

$$\psi_p^{(i)}(h_v^{(k)}) = \text{FNN}_p^{(i)}(\xi), \ \xi \sim \mathcal{N}(\mu_v, \Sigma_v), \text{ where } \mu_v = \text{FNN}_\mu(h_v^{(k)}), \Sigma_v = \text{diag}(\exp(\text{FNN}_\sigma(h_v^{(k)}))).$$

**Theorem 3.1.** For any $\epsilon > 0$, if the support of the distribution $\mathcal{P}_v^{(i)}$ lies in a bounded space of $\mathbb{R}^m$, there exists a FNN $u(\cdot) : \mathbb{R}^m \to \mathbb{R}$ (and thus its gradient $\nabla u(\cdot) : \mathbb{R}^m \to \mathbb{R}^m$) with large enough width and depth (depending on $\epsilon$) such that $\mathcal{W}_2^2(\mathcal{P}_v^{(i)}, \nabla u(\mathcal{G})) < \epsilon$ where $\nabla u(\mathcal{G})$ is the distribution generated via the mapping $\nabla u(\xi)$, $\xi \sim$ a $m$-dim non-degenerate Gaussian distribution.

A further challenge is that the Wasserstein distance between $\mathcal{P}_v^{(i)}$ and $\psi_p^{(i)}(h_v^{(k)})$ does not have a closed form. Therefore, we adopt the empirical Wasserstein distance that can provably approximate the population one as stated in Sec.8.4.1 in (Peyré et al., 2019). For every forward pass, the model will get $q$ sampled nodes $\mathcal{N}_v$, denoted by $v_1, v_2, ..., v_q$ and thus $\{h_{v_j}^{(i)}|1 \leq j \leq q\}$ are $q$ samples from $\mathcal{P}_v^{(i)}$; Next, get $q$ samples from $\mathcal{N}(\mu_v, \Sigma_v)$, denoted by $\xi_1, \xi_2, ..., \xi_q$, and thus $\{\hat{h}_v^{(i,j)} = \text{FNN}_p^{(i)}(\xi_j)|1 \leq j \leq q\}$ are $q$ samples from $\psi_p^{(i)}(h_v^{(k)})$; Adopt the following empirical surrogate loss of $\sum_{i=0}^{k-1} \mathcal{W}_2^2(\mathcal{P}_v^{(i)}, \psi_p^{(i)}(h_v^{(k)}))$ in Eq.6,

$$\min_\pi \sum_{i=0}^{k-1} \sum_{j=1}^q \|h_{v_j}^{(i)} - \hat{h}_v^{(i,\pi(j))}\|^2, \text{ s.t. } \pi \text{ is a bijective mapping:} [q] \to [q]. \tag{8}$$

Computing the above loss is based on solving a matching problem and needs the Hungarian algorithm with $O(q^3)$ complexity (Jonker & Volgenant, 1987). More efficient types of surrogate loss may be adopted, such as Chamfer loss based on greedy approximation (Fan et al., 2017) or Sinkhorn loss based on continuous relaxation (Cuturi, 2013), whose complexities are $O(q^2)$. In our experiments, as $q$ is fixed as a small constant, say 5, we use Eq.8 based on a Hungarian matching and do not introduce much computational overhead.

Although not directly related, we would like to highlight some recent works that use OT losses as distance between two graphs, where Wasserstein distance between the two sets of node embeddings of these two graphs is adopted (Xu et al., 2019a;b). Borrowing such a concept, one can view our OT-loss as also to measure the distance between the original graph and the decoded graph.

## 3.3 Further Discussion — Decoders $\psi_s$ and $\psi_d$

The decoder $\psi_s$ to reconstruct the initial features $h_v^{(0)}$ is an FNN. The decoder $\psi_d$ to reconstruct the node degree is an FNN plus exponential neuron to make the value non-negative.

$$\psi_s(h_v^{(k)}) = \text{FNN}_s(h_v^{(k)}), \quad \psi_d(h_v^{(k)}) = \exp(\text{FNN}_d(h_v^{(k)})). \tag{9}$$

In practice, the original node features $X$ could be very high-dimensional and reconstructing them directly may introduce a lot of noise into the node representations. Instead, we may first map $X$ into a latent space to initialize $H^{(0)}$. However, if $H^{(0)}$ is used in both representation learning and reconstruction, it has the risk to collapse to trivial points. Hence, we normalize $H^{(0)} = \{h_v^{(0)}|v \in V\}$ properly via pair-norm (Zhao & Akoglu, 2020) to avoid this trap as follows

$$\{h_v^{(0)}|v \in V\} = \text{pair-norm}(\{x_v W|v \in V\}), \text{ where } W \text{ is a learnable parameter matrix.} \tag{10}$$

## 4 EXPERIMENTS

We design our experiments to evaluate `NWR-GAE`, focusing on the following research questions: **RQ1**: How does `NWR-GAE` perform on structure-role-based synthetic datasets in comparison to state-of-the-art unsupervised graph embedding baselines? **RQ2**: How do `NWR-GAE` and its ablations compare to the baselines on different types of real-world graph datasets? **RQ3**: What are the impacts of the major model parameters including embedding size $d$ and sampling size $q$ on `NWR-GAE`?

### 4.1 EXPERIMENTAL SETUP

#### 4.1.1 DATASETS

**Synthetic datasets.** Following existing works on structure role oriented graph embedding (Henderson et al., 2012; Ribeiro et al., 2017; Donnat et al., 2018), we construct multiple sets of *planted structural equivalences graphs*, which are generated by regularly placing different basic shapes (*House*, *Fan*, *Star*) along a cycle of certain length. We use node degrees as the node features, and generate the structural-role labels for nodes in the graphs in the same way as (Donnat et al., 2018).

**Real-world graph Datasets.** We use a total of nine public real-world graph datasets, which roughly belong to three types, one with proximity-oriented (assortative (Liu et al., 2020)) labels, one with structure-oriented (disassortative) labels, and one with proximity-structure-mixed labels. Among them, Cora, Citeseer, Pubmed are publication networks (Namata et al., 2012); Cornell, Texas, and Wisconsin are school department webpage networks (Pei et al., 2020); Chameleon, Squirrel are page-page networks in Wikipedia (Rozemberczki et al., 2021); Actor is an actor co-filming network (Tang et al., 2009). More details of the datasets are put in Appendix B.

#### 4.1.2 BASELINES

We compare `NWR-GAE` with a total of 11 baselines, representing four types of state-of-the-art on unsupervised graph embedding, 1) Random walk based (DeepWalk, node2vec), 2) Structural role based (RoleX, struc2vec, GraphWave), 3) Graph auto-encoder based (GAE, VGAE, ARGVA), 4) Contrastive learning based (DGI, GraphCL, MVGRL).

- **DeepWalk** (Perozzi et al., 2014): a two-phase method which generates random walks from all nodes in the graph and then treats all walks as input to a Skipgram language model.
- **node2vec** (Grover & Leskovec, 2016): a method similar to DeepWalk but uses two hyper-paramters $p$ and $q$ to control the behavior of the random walker.
- **RolX** (Henderson et al., 2012): a structure role discovery method which extracts features and group features of each node, and then generates structure roles via matrix factorization.
- **struc2vec** (Ribeiro et al., 2017): a graph structural embedding method that encodes the structural similarities of nodes and generates structural context for each node.
- **GraphWave** (Donnat et al., 2018): a graph structural embedding method that models the diffusion of a spectral graph wavelet and learns embeddings from network diffusion spreads.
- **GAE** (Kipf & Welling, 2016): a GCN encoder trained with a link reconstruction decoder.
- **VGAE** (Kipf & Welling, 2016): the variational version of GAE.
- **ARGVA** (Pan et al., 2018): a method which considers topological structure and node content, trained by reconstructing the graph structure.
- **DGI** (Veličković et al., 2018b): a GCN encoder trained via graph mutual information maximization.
- **GraphCL** (You et al., 2020): a contrastive learning method via four types of graph augmentation.
- **MVGRL** (Hassani & Khasahmadi, 2020): a constastive learning method by contrasting encodings from first-order neighbors and a graph diffusion.

#### 4.1.3 EVALUATION METRICS

Following (Donnat et al., 2018), we measure the performance of compared algorithms on the synthetic datasets based on agglomerative clustering (with single linkage) with the following metrics

- **Homogeneity**: conditional entropy of ground-truth among predicting clusters.
- **Completeness**: ratio of nodes with the same ground-truth labels assigned to the same cluster.
- **Silhouette score**: intra-cluster distance vs. inter-cluster distance.

Table 1: Performance of compared algorithms on structure role identification with synthetic data.

| Dataset | Metrics | DeepWalk | node2vec | RolX | struc2vec | GraphWave | GAE | VGAE | ARGVA | DGI | GraphCL | MVGRL | NWR-GAE |
|---|---|---|---|---|---|---|---|---|---|---|---|---|---|
| House | Homogeneity | 0.01 | 0.01 | **1.0** | 0.99 | **1.0** | **1.0** | 0.25 | 0.28 | **1.0** | **1.0** | **1.0** | **1.0** |
| | Completeness | 0.01 | 0.01 | **1.0** | 0.99 | **1.0** | **1.0** | 0.27 | 0.28 | **1.0** | **1.0** | **1.0** | **1.0** |
| | Silhouette | 0.29 | 0.33 | **0.99** | 0.45 | **0.99** | **0.99** | 0.21 | 0.19 | **0.99** | **0.99** | **0.99** | **0.99** |
| House Perturbed | Homogeneity | 0.06 | 0.03 | **0.65** | 0.21 | 0.52 | 0.36 | 0.29 | 0.24 | 0.24 | 0.41 | 0.63 | **0.65** |
| | Completeness | 0.06 | 0.03 | 0.69 | 0.24 | 0.53 | 0.37 | 0.29 | 0.24 | 0.25 | 0.42 | 0.63 | **0.72** |
| | Silhouette | 0.25 | 0.28 | 0.51 | 0.18 | 0.39 | 0.67 | 0.51 | 0.44 | 0.69 | 0.69 | 0.69 | **0.94** |
| Varied | Homogeneity | 0.26 | 0.24 | 0.91 | 0.63 | 0.87 | 0.65 | 0.50 | 0.66 | 0.36 | **0.93** | 0.93 | 0.93 |
| | Completeness | 0.23 | 0.22 | 0.93 | 0.58 | 0.88 | 0.69 | 0.36 | 0.57 | 0.37 | 0.89 | 0.89 | **0.94** |
| | Silhouette | 0.35 | 0.40 | 0.82 | 0.24 | 0.66 | 0.66 | 0.21 | 0.23 | 0.94 | 0.93 | 0.90 | **0.95** |
| Perturbed & Varied | Homogeneity | 0.30 | 0.30 | 0.74 | 0.46 | 0.63 | 0.44 | 0.42 | 0.57 | 0.36 | 0.70 | 0.73 | **0.78** |
| | Completeness | 0.27 | 0.27 | 0.72 | 0.43 | 0.60 | 0.45 | 0.43 | 0.49 | 0.36 | 0.63 | 0.67 | **0.81** |
| | Silhouette | 0.33 | 0.36 | 0.61 | 0.29 | 0.50 | 0.52 | 0.21 | 0.20 | 0.45 | 0.69 | 0.61 | **0.84** |

Table 2: Performance of compared algorithms on different types of real-world graph datasets.

| | Proximity | | | Structure | | | Mixed | | |
|---|---|---|---|---|---|---|---|---|---|
| **Algorithms** | Cora | Citeseer | Pubmed | Cornell | Texas | Wisconsin | Chameleon | Squirrel | Actor |
| DeepWalk | 82.97 ± 1.67 | 68.99 ± 0.95 | 82.39 ± 4.88 | 41.21 ± 3.40 | 41.89 ± 7.81 | 43.62 ± 2.46 | 68.03 ± 2.13 | 59.22 ± 2.35 | 23.84 ± 2.14 |
| node2vec | 81.93 ± 1.43 | 64.56 ± 1.65 | 81.02 ± 1.48 | 40.54 ± 1.62 | 48.64 ± 2.92 | 36.27 ± 2.08 | 65.67 ± 2.31 | 48.29 ± 1.67 | 24.14 ± 1.02 |
| RolX | 29.70 ± 2.89 | 20.90 ± 0.72 | 39.85 ± 2.33 | 25.67 ± 11.78 | 42.56 ± 7.13 | 24.92 ± 13.43 | 22.75 ± 2.12 | 20.50 ± 1.18 | 25.42 ± 0.55 |
| struc2vec | 41.46 ± 1.49 | 51.70 ± 0.67 | 81.49 ± 0.33 | 23.72 ± 13.69 | 47.29 ± 7.21 | 24.59 ± 12.14 | 60.63 ± 2.90 | 52.59 ± 0.69 | 25.13 ± 0.79 |
| GraphWave | 28.83 ± 2.39 | 20.79 ± 1.59 | 20.96 ± 2.35 | 45.96 ± 2.20 | 37.45 ± 7.09 | 39.24 ± 5.16 | 22.03 ± 1.09 | 29.34 ± 1.12 | 27.29 ± 3.09 |
| GAE | 72.06 ± 2.54 | 57.10 ± 1.62 | 73.24 ± 0.88 | 45.40 ± 9.99 | 58.78 ± 3.41 | 34.11 ± 8.06 | 22.03 ± 1.09 | 29.34 ± 1.12 | 28.63 ± 1.05 |
| VGAE | 72.87 ± 1.48 | 60.78 ± 1.92 | 81.34 ± 0.79 | 49.32 ± 9.19 | 39.18 ± 8.96 | 38.27 ± 6.12 | 20.17 ± 1.30 | 19.57 ± 1.63 | 26.41 ± 1.07 |
| ARGVA | 72.88 ± 3.83 | 63.36 ± 2.08 | 75.32 ± 0.63 | 41.08 ± 4.85 | 43.24 ± 5.38 | 41.17 ± 5.20 | 21.17 ± 0.78 | 20.61 ± 0.73 | 28.97 ± 1.17 |
| DGI | 84.76 ± 1.39 | 71.68 ± 1.54 | **84.29 ± 1.07** | 46.48 ± 7.97 | 52.97 ± 5.64 | 55.68 ± 2.97 | 25.89 ± 1.49 | 25.89 ± 1.62 | 20.45 ± 1.32 |
| GraphCL | 84.23 ± 1.51 | 73.51 ± 1.73 | 82.59 ± 0.71 | 44.86 ± 3.73 | 46.48 ± 5.85 | 53.72 ± 1.07 | 26.27 ± 1.53 | 21.32 ± 1.66 | 28.64 ± 1.28 |
| MVGRL | **86.23 ± 2.71** | **73.81 ± 1.53** | 83.94 ± 0.75 | 53.51 ± 3.26 | 56.75 ± 5.97 | 57.25 ± 5.94 | 58.73 ± 2.03 | 40.64 ± 1.15 | **31.07 ± 0.29** |
| NWR-GAE w/o deg. | 79.75 ± 2.41 | 70.68 ± 0.51 | 81.04 ± 0.57 | 54.05 ± 5.97 | 65.13 ± 6.79 | 65.09 ± 5.04 | 70.54 ± 2.63 | 64.20 ± 1.07 | 29.57 ± 1.14 |
| NWR-GAE w/o Hun. | 82.38 ± 2.11 | 68.95 ± 1.79 | 80.30 ± 0.36 | 50.81 ± 4.44 | 67.59 ± 8.67 | 62.35 ± 4.88 | 70.98 ± 2.05 | 62.63 ± 2.27 | 29.47 ± 1.32 |
| NWR-GAE w/o dist. | 80.67 ± 2.49 | 69.78 ± 2.42 | 81.13 ± 0.27 | 49.72 ± 5.04 | 65.94 ± 6.53 | 68.12 ± 4.41 | 71.64 ± 1.85 | 62.07 ± 1.41 | 28.59 ± 0.96 |
| NWR-GAE full | 83.62 ± 1.61 | 71.45 ± 2.41 | 83.44 ± 0.92 | **58.64 ± 5.61** | **69.62 ± 6.66** | **68.23 ± 6.11** | **72.04 ± 2.59** | **64.81 ± 1.83** | 30.17 ± 0.17 |

For the experiments on real-world graph datasets, we learn the graph embeddings with different algorithms and then follow the standard setting of supervised classification. To be consistent across all datasets in our experiments, we did not follow the standard semi-supervised setting (20 labels per class for training) on Cora, Citeseer and Pubmed, but rather randomly split all datasets with 60% training set, 20% validation set, and 20% testing set, which is a common practice on WebKB and Wikipedia network datasets (i.e. Cornell, Texas, Chameleon, etc.) (Liu et al., 2020; Pei et al., 2020; Ma et al., 2021). The overall performance is the average performance over 10 random splits. Due to space limitation, we put details of the compared models and hyper-parameter settings in Appendix C.

## 4.2 Performance on Synthetic Datasets (RQ1)

We compare NWR-GAE with all baselines on the synthetic datasets. As can be observed in Table 1, NWR-GAE has the best performance over all compared algorithms regarding all evaluation metrics, corroborating its claimed advantages in capturing the neighborhood structures of nodes on graphs.

Specifically, (1) on the purely symmetrical graphs of *House*, NWR-GAE achieves almost perfect predictions on par with RolX, GraphWave and GAE; (2) on the asymmetrical graphs of *House Perturbed* (by randomly adding or deleting edges between nodes), *Varied* (by placing varied shapes around the base circle such as *Fans* and *Stars*) and *Varied perturbed*, the performances of all algorithms degrade significantly, while NWR-GAE is most robust to such perturbation and variance; (3) among the baselines, proximity oriented methods of DeepWalk and node2vec perform extremely poorly due to the ignorance of structural roles, while structural role oriented methods of RolX, strc2vec and GraphWave are much more competitive; (4) GNN-based baselines of GAE, VGAE and DGI still enjoy certain predictive power, due to the structure representativeness of their GIN encoder, but such representativeness is actually impeded by their improper loss functions for structural role identification, thus still leading to rather inferior performance.

## 4.3 Performance on Real-World Datasets (RQ2)

All 11 baselines are included in the real-world datasets performance comparison. As can be observed in Table 2, NWR-GAE achieves significantly better results compared to all baselines on all structure-oriented and proximity-structure-mixed graph datasets, and maintains close performance to the best baselines of DGI, GraphCL and MVGRL on all proximity-oriented graph datasets. Note that, most of the strong baselines can only perform well on either of the proximity-oriented or structure-oriented datasets, while NWR-GAE is the only one that performs consistently well on all three types. Such results again indicate the superiority of NWR-GAE in capturing the structural information on graphs, yet without significantly sacrificing the capturing of node proximity.

Specifically, (1) Most GNN-based methods (both auto-encoder based, and contrastive learning based), like GAE, VGAE, ARGVA, DGI and GraphCL only perform well on graphs with proximity-oriented labels due to their strongly proximity-preserving loss functions, and perform rather poorly on most graphs with structure-oriented and mixed labels; MVGRL achieves relatively better but still inferior performance to ours on structure oriented datasets due to its structure aware contrastive training. (2) the random walk based methods of DeepWalk and node2vec perform well mostly on graphs with proximity-oriented and mixed labels; (3) the structural role oriented methods of RolX, struc2vec and GraphWave perform rather poorly on graphs with proximity-oriented and mixed labels, and sometimes also the structure-oriented ones, due to their incapability of modeling node features.

### 4.4    IN-DEPTH ANALYSIS OF `NWR-GAE` (RQ3)

In Table 2, we also compare several important ablations of `NWR-GAE`, i.e., the one without degree predictor, the one without Hungarian loss (replaced by a greedy-matching based loss), and the one without feature distribution reconstructor, which all lead to inferior performance on all datasets, demonstrating the effectiveness of our novel model designs.

In Figure 4, we present results on hyper-parameter and efficiency studies of `NWR-GAE`. Embedding size efficiency is an important metric for unsupervised representation learning, which reflects the effectiveness in information compression. As can be seen in Figure 4 (a), `NWR-GAE` quickly converges when the embedding size reaches 200-300 on the mid-scale datasets of Chameleon and Squirrel, and maintains good performance as the sizes further increase without overfitting. Moreover, as we can observe from the training curves of `NWR-GAE` on Chameleon in Figure 4 (b), the performance of `NWR-GAE` is robust to the neighborhood sampling sizes, where the larger size of 15 only leads to slightly better performance than the smaller size of 5. `NWR-GAE` also converges rapidly after 1-2 training epochs, achieving significantly better performance compared with the best baseline of DeepWalk and typical baselines like DGI. As we can observe from the runtimes of `NWR-GAE` on Chameleon in Figure 4 (c), smaller neighborhood sampling sizes do lead to significantly shorter runtimes, while our runtimes are much better compared with typical baselines like DeepWalk and DGI, especially under the fact that we can converge rather rapidly observed from Figure 4 (b).

Finally, to further understand the effectiveness of our proposed neighborhood Wasserstein reconstruction method, we provide detailed results on how well the generated neighbor features can approximate the real ones in Appendix D. The results show that the method can indeed recover the entire real neighbor features, and the approximation gets better as we use a larger sampling factor $q$ especially on larger datasets, while the performances on downstream tasks are already satisfactory even with smaller $q$ values, allowing the training to be rather efficient.

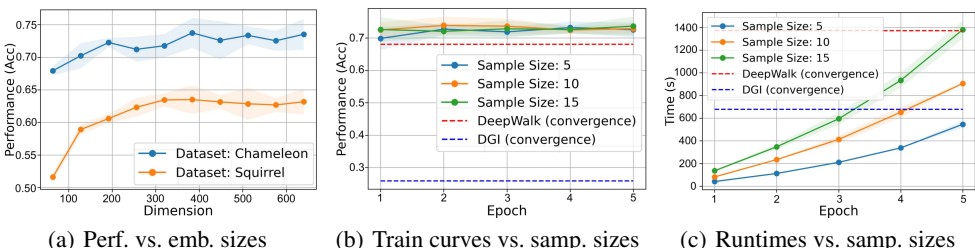

| (a) Perf. vs. emb. sizes | (b) Train curves vs. samp. sizes | (c) Runtimes vs. samp. sizes |

Figure 4: Hyper-parameter and efficiency studies of `NWR-GAE`

## 5    CONCLUSION

In this work, we address the limitations of existing unsupervised graph representation methods and propose the first model that can properly capture the proximity, structure and feature information of nodes in graphs, and discriminatively encode them in the low-dimensional embedding space. The model is extensively tested on both synthetic and real-world benchmark datasets, and the results strongly support its claimed advantages. Since it is general, efficient, and also conceptually well-understood, we believe it to have the potential to serve as the de facto method for unsupervised graph representation learning. In the future, it will be promising to see its applications studied in different domains, as well as its potential issues such as robustness and privacy carefully analyzed.

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

# A  PROOF OF THEOREM 3.1

Theorem 3.1 can be obtained by properly revising Theorem 2.1 in (Lu & Lu, 2020). We first state Theorem 2.1 in (Lu & Lu, 2020).

**Theorem A.1** (Theorem 2.1 in Lu & Lu (2020))**.** Let $P$ and $Q$ be the target and the source distributions respectively, both defined on $\mathbb{R}^m$. Assume that $\mathbb{E}_{x \sim P}\|x\|^3$ is bounded and $Q$ is absolutely continuous with respect to the Lebesgue measure. It holds that for any given approximation error $\epsilon$, setting $n = O(\frac{1}{\epsilon^m})$, there is a fully connected and feed-forward deep neural network $u(\cdot)$ of depth $L = \lceil \log_2 n \rceil$ and width $N = 2L$, with $d$ inputs and a single output and with ReLU activation such that $\mathcal{W}_1(P, \nabla u(Q)) < \epsilon$. Here, $\nabla u(\cdot)$ is the function $\mathbb{R}^d \to \mathbb{R}^d$ induced by the gradient of $u$ while $\nabla u(Q)$ is the distribution that is generated from the distribution $Q$ through the mapping $\nabla u(\cdot)$.

To prove Theorem 3.1, we just need to verify the conditions in the above theorem. We need to show that $P = \mathcal{P}_v^{(i)}$ has a bounded $\mathbb{E}_{x \sim P}\|x\|^3$, and $Q$ as a non-degenerate $m$-dim Gaussian distribution is absolutely continuous with respect to the Lebesgue measure. Also, the original statement is for $\mathcal{W}_1(\cdot, \cdot)$ while we use $\mathcal{W}_2(\cdot, \cdot)$. We need to show the connection between $\mathcal{W}_1(\cdot, \cdot)$ and $\mathcal{W}_2(\cdot, \cdot)$. Fortunately, all these aspects are easy to be verified.

$P = \mathcal{P}_v^{(i)}$ has a bounded 3-order moment because the support $\mathcal{P}_v^{(i)}$ are in a bounded space of $\mathbb{R}^m$.

$Q$ as a Gaussian distribution is absolutely continuous with respect to the Lebesgue measure, as long as $Q$ is not degenerate.

Lastly we show the connection between $\mathcal{W}_1(\cdot, \cdot)$ and $\mathcal{W}_2(\cdot, \cdot)$. The support $P = \mathcal{P}_v^{(i)}$ is bounded, i.e., $\delta \in \text{supp}(P)$, $\|\delta\|_2 \leq B$. According to Lu & Lu (2020), the $Q = \nabla u(\mathcal{G})$ also has bounded support. Without loss of generality, $\delta \in \text{supp}(Q)$, $\|\delta\|_2 \leq B$. Then, we may show that

$$
\begin{aligned}
\mathcal{W}_2^2(P, Q) &= \inf_{\gamma \in \Gamma(P,Q)} \left[ \int_{\mathcal{Z} \times \mathcal{Z}'} \|Z - Z'\|_2^2 d\gamma(Z, Z') \right] \\
&\leq 2B \inf_{\gamma \in \Gamma(P,Q)} \left[ \int_{\mathcal{Z} \times \mathcal{Z}'} \|Z - Z'\|_2 d\gamma(Z, Z') \right] \\
&\leq 2B\sqrt{m} \inf_{\gamma \in \Gamma(P,Q)} \left[ \int_{\mathcal{Z} \times \mathcal{Z}'} \|Z - Z'\|_1 d\gamma(Z, Z') \right] \\
&= 2B\sqrt{m} \mathcal{W}_1(P, Q) < 2B\sqrt{m}\epsilon.
\end{aligned}
$$

As $B$ and $m$ are just constant, so we may have $\mathcal{W}_2^2(P, Q) = O(\epsilon)$. Note that the above first inequality is because $\|Z - Z'\|_2 \leq \|Z\|_2 + \|Z'\|_2 = 2B$. The second inequality is because

$$
\begin{aligned}
&\sqrt{m} \inf_{\gamma \in \Gamma(P,Q)} \left[ \int_{\mathcal{Z} \times \mathcal{Z}'} \|Z - Z'\|_1 d\gamma(Z, Z') \right] \\
&= \lim_{i \to \infty} \left[ \int_{\mathcal{Z} \times \mathcal{Z}'} \sqrt{m} \|Z - Z'\|_1 d\gamma_i(Z, Z') \right] \\
&\qquad \text{(There exists a sequence of measures } \{\gamma_i\}_{i=1}^{\infty} \text{ achieving the infimum)} \\
&\geq \lim_{i \to \infty} \left[ \int_{\mathcal{Z} \times \mathcal{Z}'} \|Z - Z'\|_2 d\gamma_i(Z, Z') \right] \\
&\geq \inf_{\gamma \in \Gamma(P,Q)} \left[ \int_{\mathcal{Z} \times \mathcal{Z}'} \|Z - Z'\|_2 d\gamma(Z, Z') \right]
\end{aligned}
$$

# B  ADDITIONAL DATASET DETAILS

In this section, we provide some additional important details about the synthetic and real-world network datasets we use in the node classification experiments.

From table B, the node labels shows how the node are classified. For example, the node labels of citation networks are academic topics, which is highly related to proximity information. But the identity roles of website is more structural oriented since similar identity have similar hyperlinks structure. Detailed statistics of the real-world graph datasets are provided in Table 4.

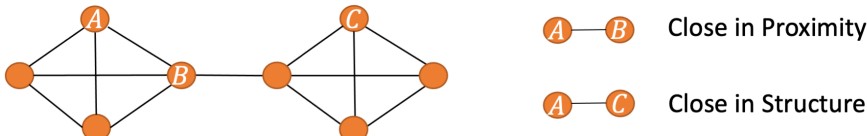

Figure 5: Toy example of proximity and structure information: A and B are "close in proximity" since they are relatively close in terms of node distances in the global network, whereas A and C are "close in structure" since they have relatively similar local neighborhood structures.

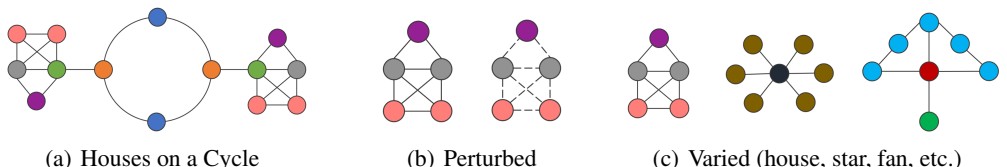

(a) Houses on a Cycle      (b) Perturbed      (c) Varied (house, star, fan, etc.)

Figure 6: Synthetic dataset examples (Colors denote structural role labels and dashed lines denote randomly adding or deleting edges)

## C   DETAILED SETTINGS OF THE COMPARED MODELS

**DeepWalk**: for all datasets, we ran 10 random walks of length 40 and input these walks into a skipgram langauge model. We implement the DeepWalk method based on Python code (GPLv3 license) provided by Perozzi et al. (2014).

**node2vec**: similar to DeepWalk, we also ran 10 random walks of length 40 with p and q equal to 0.25. The implementation is provided by Grover & Leskovec (2016).

**RolX**: by using Henderson et al. (2012)'s code, we set the initial learning rate as 0.01 and ran the algorithm with batch size 32.

**struc2vec**: similar to DeepWalk and node2vec, we ran the same number of random walks with walk length 40. We set window size 5 as context size for optimization. The reference implementation is provided by Ribeiro et al. (2017)

**GraphWave**: the implementation is under the MIT license and provided by Stanford SNAP lab (Donnat et al., 2018). All the parameters are automatically determined by the algorithm itself.

**GAE**: we use a two layer GCN as graph encoder for node classification task and a three layer GIN for synthetic structural identification task. The decoder hyperparameters are selected by cross-validation, the implementation is based on the code provided by Kipf & Welling (2016)

**VGAE**: same encoder structure and similar decoder as GAE, the implementation is also based on Kipf & Welling (2016).

**ARGVA**: same encoder structure as GAE and VGAE. We use the Tensorflow implementation based on the implementation of Pan et al. (2018).

**DGI**: same encoder structure as GAE and VGAE, the initial number of epochs is 10000 but may early stop when the algorithm is converged. We use the DGI method based on the implementation of Veličković et al. (2018b).

Table 3: Real-world network datasets description

| Datasets | Type | Nodes | Edges | Node Features | Node Labels |
|---|---|---|---|---|---|
| Cora Citeseer Pubmed | Citation Networks | Papers | Citations | Bag-of-words of papers | Academic topics |
| Cornell Texas Wisconsin | WebKB | Web pages | Hyperlinks | Bag-of-words of web pages | Identity roles (Student, Project, etc.) |
| Chameleon Squirrel | Wikipedia Network | Web pages | Mutual links | Informative nouns | Number of monthly traffic |

Table 4: Statistics of the real-world graph datasets.

| Task type | Proximity | | | Structure | | | Mixed | | |
|---|---|---|---|---|---|---|---|---|---|
| **Dataset** | Cora | Citeseer | Pubmed | Texas | Wisconsin | Cornell | Chameleon | Squirrel | Actor |
| Nodes | 2708 | 3327 | 19717 | 183 | 183 | 251 | 2277 | 5201 | 7600 |
| Edges | 5429 | 4732 | 44338 | 295 | 309 | 499 | 36101 | 217073 | 33544 |
| Features | 1433 | 3703 | 500 | 1703 | 1703 | 1703 | 2325 | 2089 | 931 |
| Classes | 7 | 6 | 3 | 5 | 5 | 5 | 5 | 5 | 5 |

**GraphCL**: use the default encoder GraphSage and all the four different types of augumentation. The implementation is based on You et al. (2020).

**MVGRL**: the same default GCN encoder as GAE, We use the PyTorch implementation of Hassani & Khasahmadi (2020).

**NWR-GAE**: same encoder structure as the above GNN-based methods, learning rate, trade-off weight between degree loss and neighborhood reconstruction loss and sample size $q$ are tuned using grid search.

**Encoder setting.** We used the same GCN encoders for almost all baselines because our contribution in this work is on the decoder and we wanted to be fair and focus on the comparison of different decoders. Specifically, we use the default encoders in their original work for all baselines that involve GNN encoders (GraphSAGE (Hamilton et al., 2017a) for GraphCL (You et al., 2020), GCN (Kipf & Welling, 2017) for GAE (Kipf & Welling, 2016), VGAE (Kipf & Welling, 2016), DGI (Veličković et al., 2018b), MVGRL (You et al., 2020)). The GCN encoders all have the same two-layer architecture, which is the same as we used for `NWR-GAE`. In our experiments, we have actually tried to use other GNNs as the encoders for all methods including `NWR-GAE` and the baselines, but found GCN to have the best performance. For example, in the following Table C, we provide the results of GAE with the GIN (Xu et al., 2019c) and GraphSage (Hamilton et al., 2017a) encoders, which are clearly worse than that with GCN.

**Hyper-parameter tuning.** For all compared models, we performed hyper-parameter selection on learning rate {5e-3, 5e-4, 5e-5, 5e-6, 5e-7} and epoch size {100, 200, 300, 400, 500, 600}. For `NWR-GAE`, we selected the sample size $q$ from {3, 5, 8, 10}, and the trade-off weight parameters $\lambda_d$, $\lambda_s$ from {10, 1, 1e-1, 1e-2, 1e-3, 1e-4, 1e-5}.

**Notes on fairness.** We set all random walk based method (DeepWalk, node2vec, struc2vec) with same number of random walks and walk lengths. For all graph neural network based methods (GAE, VGAE, DGI, `NWR-GAE`), we use the same GCN encoder with same number of encoder layer from DGL library (Wang et al., 2019). For the node classification task, all methods use the same 4-layer MLP predictor with learning rate 0.01. For all settings, we use Adam optimizer and backward propagation from PyTorch Python package, and a fix dimension size as same as the graph node feature size. Most experiments are performed on a 8GB NVIDIA GeForce RTX 3070 GPU.

Table 5: Selected baseline with different encoders

| Algorithms | Cornell | Texas | Wisconsin | Chameleon | Squirrel | Actor |
|---|---|---|---|---|---|---|
| GAE w/GCN | $45.40 \pm 9.99$ | $58.78 \pm 3.41$ | $34.11 \pm 8.06$ | $22.03 \pm 1.09$ | $29.34 \pm 1.12$ | $28.63 \pm 1.05$ |
| GAE w/GIN | $39.33 \pm 4.17$ | $53.30 \pm 2.75$ | $32.84 \pm 3.94$ | $24.99 \pm 2.56$ | $23.46 \pm 1.04$ | $25.34 \pm 1.43$ |
| GAE w/GraphSage | $44.97 \pm 4.61$ | $47.29 \pm 5.43$ | $37.25 \pm 6.25$ | $22.97 \pm 1.29$ | $20.05 \pm 0.34$ | $28.34 \pm 0.51$ |

**Notes on neighborhood sampling.** due to the heavy tailed nature of node distribution, we down sample the number of neighbors in `NWR-GAE` method. We only sampled at most 10 neighbors per node. This provides a good trade-off between efficiency and performance, which makes the method have reasonable running time on one GPU machine and also achieve great enough accuracy.

# D ADDITIONAL EXPERIMENTAL RESULTS

To further understand the effectiveness of our proposed neighborhood Wasserstein reconstruction method, we provide detailed results on how well the generated neighbor features can approximate the real ones.

Specifically, for every node v, to reconstruct k-hops neighborhood, the model will sample $q$ neighbor nodes from $\mathcal{N}_v$ in layer i, where $0 \leq i \leq k$, thus $\{h_{v_j}^{(i)}|1 \leq j \leq q\}$ denotes the q ground truth embeddings of v's neighbor. Next the model gets $q$ samples $\{\xi_j, 1 \leq j \leq q\}$ from Gaussian

distribution parameterized by GNN encoder, and uses an FNN-based transformation $\{\hat{h}_v^{(i,j)} = \text{FNN}^{(i)}(\xi_j)\}$ to reconstruct $h_{v_j}^{(i)}$.

Based on above, we create x-y pairs with

$$x = \frac{\sum_{j=1}^q (||\hat{h}_v^{(k,j)}||^2)}{q},$$

$$y = \frac{\min_{\pi:[q] \to [q]} \sum_{i=0}^{k-1} \sum_{j=1}^q ||h_{v_j}^{(i)} - \hat{h}_v^{(i,\pi(j))}||^2}{q \cdot k}.$$

Table 6: Box-plot like approximation power table.

| Dataset | 5% | 25% | 50% | 75% | 95% | q |
|---------|------|------|------|------|------|------|
| Cornell | 0.0086 | **0.0336** | 0.0618 | 0.0822 | **0.1095** | 5 |
|         | 0.0089 | 0.0344 | **0.0617** | **0.0813** | 0.1130 | 15 |
|         | **0.0027** | 0.0398 | 0.0848 | 0.1204 | 0.1436 | 30 |
| Texas | **0.0025** | 0.0477 | 0.0869 | 0.1304 | 0.1792 | 5 |
|       | 0.0078 | 0.0559 | 0.0897 | 0.1445 | 0.1828 | 15 |
|       | 0.0077 | **0.0285** | **0.0489** | **0.0869** | **0.1048** | 30 |
| Wisconsin | **0.0060** | **0.0472** | 0.0723 | 0.0966 | 0.1451 | 5 |
|           | 0.0364 | 0.0526 | **0.0671** | **0.0848** | **0.1080** | 15 |
|           | 0.0124 | 0.0529 | 0.0716 | 0.0946 | 0.1398 | 30 |
| Chameleon | 0.1590 | 0.2587 | 0.3183 | 0.3858 | 0.5067 | 5 |
|           | 0.0958 | 0.1180 | 0.1504 | 0.2120 | 0.2704 | 15 |
|           | **0.0423** | **0.0624** | **0.0884** | **0.1384** | **0.2272** | 30 |

In Table 6, we ranked all x-y pairs by the ratio , and presented the ratio at 5%, 25%, 50%, 75%, 95% (like a box-plot). As we can observe, our model can get a reasonably good approximation to the ground-truth neighborhood information (the smaller the better approximation). We tried different sample sizes $q$ such as 5, 15, 30, which do not make a large difference in the first three datasets since most nodes in the graphs have rather small numbers of neighbors. Our model shows better approximation power on the neighborhood information with larger sample sizes such as Chameleon with denser links. However, the performances on downstream tasks are already satisfactory even with smaller $q$ values (the results we show in Table 2 are with $q = 5$, and the results we show in Figure 4 (b) are with $q = 5, 10, 15$). This indicates that while our model has the ability to more ideally approximate the entire neighborhood information, it often may not be necessary to guarantee satisfactory performance in downstream tasks, which allows us to train the model with small sample sizes and achieve good efficiency.

To better understand the effect of number of GNN layers $k$ in our proposed model, we vary $k$ from 1 to 5 on the synthetic dataset of "House". The model performance only gets better when $k$ grows over 3, because the important graph structures there cannot be captured by GNNs with less than 3 layers. Motivated by such observations and understandings, we set $k$ to 4 for all real-world experiments, which can empirically capture most important local graph structures, to achieve a good trade-off between model effectiveness and efficiency.

Table 7: The effect of $k$ (exemplified on the synthetic dataset of "House").

| Metrics | k=1 | k=2 | k=3 | k=4 | k=5 |
|---------|-----|-----|-----|-----|-----|
| Homogeneity | 0.89 | 0.94 | 0.99 | **1.0** | **1.0** |
| Completeness | 0.92 | 0.93 | **1.0** | **1.0** | **1.0** |
| Silhouette | 0.88 | 0.87 | 0.91 | **0.99** | 0.98 |

To gain more insight into the behavior of different unsupervised graph embedding methods, we visualize the embedding space learned by several representative methods on Chameleon in Figure 7 (reduced to two-dimensional via PCA). As we can observe, in these unsupervised two-dimensional embedding spaces, our baselines of node2vec, struc2vec, GraphWave, GAE and DGI can hardly distinguish the actual node classes, whereas `NWR-GAE` can achieve relatively better class separation. Note that since all algorithms are unsupervised and the visualized embedding space is very low-dimensional, it is hard for any algorithm to achieve perfectly clear class separation, but `NWR-GAE`

can capture more discriminative information and deliver less uniform node distribution, which is more useful for the node classification task as consistent with our results in Table 2.

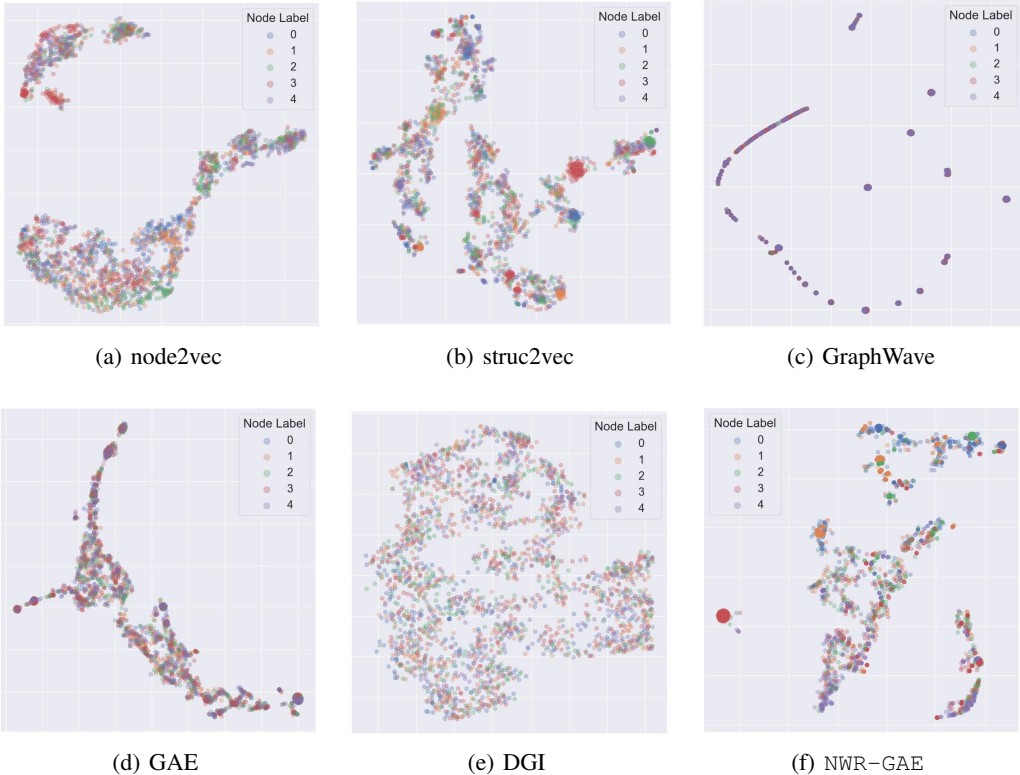

Figure 7: Embedding spaces reduced to two-dimensional through PCA on Chameleon dataset.

