# OpenReview forum: "Graph Auto-Encoder via Neighborhood Wasserstein Reconstruction"
_ICLR.cc/2022/Conference — ICLR 2022 Poster_

### Official Review · Reviewer_dc6B · 2021-10-31

**Correctness:** 4
**Technical Novelty And Significance:** 2
**Empirical Novelty And Significance:** 2
**Recommendation:** 5
**Confidence:** 4

**Main Review:**

The appreciated side of this paper is the correctness of the method, detailed illustration of implementation and the thorough experiment comparison. The proposal to adopt degree decoder, Wasserstein distance and its approximation into GAE looks reasonable to me, and the empirical examination verifies this.

The disadvantage is on the novelty side, considering that i) enforcing awareness of the context of nodes to highlight structure information is not new in the graph field (https://arxiv.org/abs/1905.12265, etc); ii) the employment of the OT theory into networks existed in even a more fancy way (https://arxiv.org/pdf/2003.03892.pdf, https://openreview.net/pdf?id=ATUh28lnSuW, etc). The benefits of individual components were established and therefore directly combining them together further boosting is not surprising.

I would recommend digging further into either theory or ablations to strengthen this interesting work, to characterize the real "structure information" captured by the proposed method compared with literature, probably following the thought as: 1) defining structure information (SI) and its quantitative assessment metric; 2) showing the proposed method can capture certain SI but the existing methods cant; 3) showing OT can capture better.

**Summary Of The Paper:**

The paper proposes a new loss in graph auto-encoder (GAE) model for unsupervised learning, composing of degree prediction and Wasserstein distance, which helps to identify structure information better vs the original loss. Extensive experiments demonstrate the advantages of the proposed loss.

**Summary Of The Review:**

I am satisfied with the solidness of the paper, including the methods part and experiments, while I feel it is limited in novelty.

---

> ### Author Response · Authors · 2021-11-16
> **Authors' initial feedback to reviewer dc6B**
>
> We thank reviewer dc6B for the high-level suggestions. However,
>
> (1) we are the first to study structure information/OT  towards unsupervised training of graph autoencoders, while the cited work https://arxiv.org/abs/1905.12265 and https://arxiv.org/pdf/2003.03892.pdf used them in totally different setting of gnn pre-training and graph sketching, letting alone how differently the exact techniques are designed. The third cited work https://openreview.net/pdf?id=ATUh28lnSuW is actually this work itself;
>
> (2) The recommended procedures are exactly what we are doing in this work: we (i) defined structure information as the neighborhood distribution, (ii) proposed NWR as an effective tool to approximate such distribution and (iii) argued based on OT theory to show why we can capture the distribution better.

---

### Official Review · Reviewer_mWmk · 2021-11-01

**Correctness:** 4
**Technical Novelty And Significance:** 3
**Empirical Novelty And Significance:** 3
**Recommendation:** 6
**Confidence:** 3

**Main Review:**

Strengths:
1. The method is intuitive and the way that the neighbourhood information is reconstructed appears novel.
2. The empirical evaluation is extensive and highlights the models benefit across a range of different datasets when compared to several categories of baseline approaches, covering both structure-based, graph auto-encoder based and contrastive learning approaches.
3. The paper is mostly well written.

Concerns:
1. How does an increase in k affect the model?
2. Delta in the expression for P_v^(0) does not seem to be defined, which impacts the clarity of how the distribution is computed.
3. A loss is introduced to predict the degree based on the node feature, however, it is not explicitly used in the neighbourhood reconstruction process. Why not sample q according to or proportional to the degree? How is the neighbour-sampling handled when a node has less than q neighbours?

Minor:
- For improved clarity, I suggest to include in Table 2, the heading that indicates which datasets are structure-oriented or proximity-oriented, etc. (as is done in Table 4 in the appendix).
- The second sentence in Section 3.2 seems to be incomplete.
- In the second sentence of Section 4.3, you mention that DGI is the best performing model. Should it be MVGRL?

**Summary Of The Paper:**

The paper proposes a novel approach to graph representation learning. In particular, a graph auto-encoder is proposed that aims to better capture the topological structure by utilising a neighbourhood reconstruction and a degree reconstruction objective. An optimal-transport based objective is proposed for the neighbourhood reconstruction that optimises the 2-Wasserstein distance between the decoded distribution and an empirical estimate of the neighbourhood distribution. An extensive experimental analysis is performed, highlighting the benefits of the proposed approach on a range of synthetic datasets to capture structure information. The experimental results also highlight its robustness across 9 different real-world graph datasets (ranging from proximity-oriented to structure-oriented datasets).

**Summary Of The Review:**

Overall, the paper is well written and presents an interesting and efficient approach to graph representation learning. I lean towards accepting the paper if the authors address the above-mentioned concerns and questions.

---

> ### Author Response · Authors · 2021-11-16
> **Authors' initial feedback to reviewer mWmk**
>
> We thank reviewer mWmk for the positive evaluations and insightful questions.
>
> (1) We selected the $k$ value as 4 in our experiments, based on our experiments on the synthetic datasets. Specifically, to capture certain graph structures that require at least $k$-hop information, the GNN encoder has to be no shallower than $k$ layers. For example, in the following table, we showcase the performance of our model on the synthetic data of “House”, with varying $k$ values, where the performance only gets good when $k$ grows over $3$. We have also included this in Appendix D in our newly submitted PDF. Since most important structures are not too complex, and deeper GNNs lead to higher computational complexity, we believe $k=4$ is generally a good choice, and used it for all real-data experiments;
>
> (2) $\delta$ in the expression of $P_v^{(0)}$ is the dirac delta function. $\delta_{x=x_0}$ is 0 everywhere except when $x=x_0$ while the integration of $\delta_{x=x_0}$ over all $x$ is 1. Dirac delta function is often used to represent a discrete distribution in a continuous space;
>
> (3) We use a degree generator and a feature generator to jointly capturing neighborhood information, regarding neighborhood size and feature distribution, in a parallel and decoupled way, so there is no clear advantage in deeply coupling the two. Moreover, using a fixed sample size $q$ is good for the ease of implementation, which is a common practice in neighborhood sampling for GNNs;
>
> (4) We appreciate all other constructive suggestions on improving the presentation of our work, and we have followed all of them in the newly submitted PDF.
>
> | Metrics      | k=1  | k=2  | k=3  | k=4  | k=5  |
> |--------------|------|------|------|------|------|
> | Homogeneity  | 0.89 | 0.94 | 0.99 | 1.0  | 1.0  |
> | Completeness | 0.92 | 0.93 | 1.0  | 1.0  | 1.0  |
> | Silhouette   | 0.88 | 0.87 | 0.91 | 0.99 | 0.98 |

---

> ### Author Response · Authors · 2021-12-08
> **Looking forward to your further comments**
>
> We thank the reviewer for the time to check our manuscript and read our response.
>
> We are wondering if your concerns have been resolved by our response. We are looking forward to your further comments.

---

### Official Review · Reviewer_hC5R · 2021-11-02

**Correctness:** 3
**Technical Novelty And Significance:** 3
**Empirical Novelty And Significance:** 3
**Recommendation:** 6
**Confidence:** 4

**Main Review:**


**Writing**
1. While most of the paper is well written, it can be still improved. For instance, the paper mentions that existing approaches are either structure-oriented or proximity-oriented approaches and they cannot distinguish certain node pairs. However, the concept of the structure or proximity information of a graph is not defined and introduced well at all in the introduction, making the paper hard to follow at the very beginning.
2. Figure 1 tries to illustrate the disadvantages of existing approaches. It is unclear why there are two nodes with the same numbers (such as 5 and 4). Are they the node labels or IDs in the graph?

**Method**

  The time complexity of the proposed method looks very high. The paper briefly describes the time complexity of Eq 8. It would be good to know the overall time complexity of the proposed algorithm.


**Experiments**

1. The idea of training a graph encoder in supervised manner links to the early graph embedding and the recent self-supervised learning (SSL) for graph data. It is good to see that both random walk based approaches, as well as SSL based approaches, are compared in the experiments. But it is unclear why GraphCL and MVGRL are only compared on the real-life datasets but are missing in the synthetic datasets in Table 1.
2. The proposed algorithm is significantly worse than the SSL based approaches such as DGL and MVGRL on Cora, Citeseer, and Pubmed. These datasets are identified as *Proximity*-oriented datasets in Table 2. More detailed explanation would be expected.
3. All datasets are relatively smaller in this paper.  It is unclear the scalability of the proposed method.

**Summary Of The Paper:**

This paper studies the problem of graph representation learning with graph autoencoder. The paper argues that most GNNs are designed for semi-supervised learning and cannot learn task-agnostic embedding. As a result, the paper proposes a graph autoencoder architecture that trains the GNN in an unsupervised manner. The key idea is to develop a decoder to reconstruct both the node degree and feature distribution. Experimental results show that the results outperform existing autoencoder baselines in several datasets.

**Summary Of The Review:**

The paper describes a new graph autoencoder approach that could encoder more information into the latent space. However, the scalability of the proposed method is questionable. Also, the proposed method did not outperform baselines on three datasets.

---

> ### Author Response · Authors · 2021-11-16
> **Authors' initial feedback to reviewer hC5R**
>
> We thank reviewer hC5R for the positive evaluations, insightful questions and constructive suggestions.
>
> Regarding writing, (1) in our newly submitted PDF, we have added an illustrative example in the Appendix B to clearly define structure information and proximity information, and we have added a discussion and pointer to this example in the Introduction section; (2) In Figure 1, we intentionally put two 4’s and two 5’s, which denote nodes that are indistinguishable in all three perspectives (feature, proximity, structure), so an unsupervised embedding algorithm should basically embed them as the same. We will add explicit discussions about this in a further revision.
>
> Regarding method, the reviewer is correct that the method will have a pretty high complexity (O($Nd^3$), with $N$ as the number of nodes and $d$ as the average node degree), if full reconstruction/matching of the neighborhoods are attempted. However, we are aware of this drawback and proposed a neighborhood sampling method as discussed under Theorem 3.1 in the paper, with a fixed sampling size $q$, which we set to be a small number of 5 for all results in Table 2, and conducted experiments on the impact of it in Table 6 in Appendix D. With $q$ fixed as a small number, the overall complexity of our method is effectively O($N$), and the data become regular and thus convenient for batch computation on GPU. The actual runtime of our method is on par with most existing unsupervised GNNs in our experiments, as shown in Figure 4(c) in the paper.
>
> Regarding experiments, (1) in our newly submitted PDF, we have added experimental results with ARGVA, GraphCL and MVGRL on the synthetic datasets in Table 1; (2) It is not our intention to claim that our method (or any method) can achieve superior performance in any datasets. The baselines like DGI and MVGRL perform very well on the commonly used Cora, Citeseer and Pubmed datasets, and we believe this is mainly because they have applied significant tuning of their model designs to “overfit” such datasets. Such strengths clearly do not fully transfer to other datasets as we showed in the experiments; (3) As we discussed above, while we focus on the effectiveness in this paper, the scalability of our method is conceptually on par with existing unsupervised GNNs. Moreover, the datasets we use include some relatively large ones such as Pubmed with 19K nodes and 44K links and Actor with 7K nodes and 33K links, which are on par with the sizes of datasets used in most GNN studies.

---

> ### Author Response · Authors · 2021-12-08
> **Looking forward to your further comments**
>
> We thank the reviewer for the time to check our manuscript and read our response.
>
> We are wondering if your concerns have been resolved by our response. We are looking forward to your further comments.

---

### Official Review · Reviewer_W8Ew · 2021-11-03

**Correctness:** 3
**Technical Novelty And Significance:** 2
**Empirical Novelty And Significance:** 3
**Recommendation:** 8
**Confidence:** 4

**Main Review:**

Strengths: Especially clear descriptive figures and writing. Experimental benefits on both toy and real world datasets and additional exploration as to why these improvements might be occurring.

Well motivated and well placed in the literature, the empirical comparisons are extensive.

The idea of NWR is seems novel and useful to me.

Weaknesses:

The notation is a bit cumbersome with many subscripts and superscripts that are difficult to keep track of.

Some of the choices in matching neighborhoods seem a bit arbitrary and not sufficiently justified (see questions below), although this is somewhat understandable as there are large number of tricks used in this paper.

Limited theoretical contribution, with some small inconsistencies.

Questions:
How much does the neighborhood Wasserstein reconstruction improve over a naive reconstruction of something like the mean of the neighborhood? Is it the Wasserstein distribution matching that is important, or just that it helps to reconstruct something about the neighborhood of each node.

Theorem 3.1 seems a bit out of place given that you don't actually want a universally approximating network here, if you had a large enough network then the NWR loss would have no benefit right, as the FNN \psi_p^(i) could always approximate the neighborhood distribution from any initial \mu, \sigma input?

Do you use W_2^2 (as in eq. 6, Figure 3) or W_2 ( as in eq. 1)? Not that it matters much but would be good to clarify.

Small notes:
It would be slightly better to prove approximation in the W_2 metric rather than W_1 in Theorem 3.1, as this is what is used in the empirical results and the rest of the paper. This seems trivially true to me.

Theorem 3.1 You are missing a closing paren in W_1(P, u(G))

**Summary Of The Paper:**

The authors propose a graph autoencoder architecture using what they term as Neighborhood Wasserstein Reconstruction (NWR). They show experimentally that this reconstruction loss improves the embedding performance in structure-oriented graph mining tasks.


**Summary Of The Review:**

This paper was an interesting read and provided insightful and clear justifications for the results. There remain some questions as to the theory and some particular loss choices made, but these are relatively minor points.

---

> ### Author Response · Authors · 2021-11-16
> **Authors' initial feedback to reviewer W8Ew**
>
> We thank reviewer W8Ew for the positive evaluation and insightful questions. In a revision, we will try to simplify the notations, though we believe some of them are necessary to guarantee our model is correctly defined. Regarding your specific questions about our model designs and theoretical justifications, we would like to highlight that
>
> (1) The usage of NWR (Neighborhood Wasserstein Reconstruction) is based on our understanding of neighborhood structural information as a distribution of neighbor embeddings, which has discrete support in a continuous space. Wasserstein distance is a well-defined metric if we want the model to learn and approximate such distributions. Theoretically, reconstructions just based on the mean of the neighborhood may lose a lot of structural information. Empirically, the experiments over the synthetic datasets can easily show the failure of using just the mean.
>
> (2) Theorem 3.1 is to guarantee that our strategy to approximate the neighborhood distribution in Wasserstein distance by using an FNN-transformed Gaussian is theoretically sound (from the perspective of model capacity). This is not trivially true. Note that our $\mu$ and $\sigma$ are not arbitrary. They depend on the latent feature representation. This is crucial. Theorem 3.1 just shows that our proposed decoder has the expressive power to decode the structural information from the latent features via the computation pipeline：latent feature -> $\mu$, $\sigma$ -> neighborhood distribution. Theorem 3.1 is important to guarantee our designed decoder is valid.
>
> (3) We use the 2-Wasserstein $\mathcal{W_2^2}$ to compute the loss, while Eq 1 defines the general Wasserstein loss.
>
> (4) We have fixed the statement in Theorem 3.1 by using approximation in the $\mathcal{W}_2$ metric rather than $\mathcal{W}_1$, and updated the proof in Appendix A in our newly submitted PDF.

---

> ### Author Response · Authors · 2021-12-08
> **Looking forward to your further comments**
>
> We thank the reviewer for the time to check our manuscript and read our response again.
>
> We are wondering if your concerns have been resolved by our response. We are looking forward to your further comments.

---

### Decision · Program_Chairs · 2022-01-20

**Decision:**

Accept (Poster)

**Comment:**

The paper proposes a novel approach to graph representation learning. In particular, a graph auto-encoder is proposed that aims to better capture the topological structure by utilising a neighbourhood reconstruction and a degree reconstruction objective. An optimal-transport based objective is proposed for the neighbourhood reconstruction that optimises the 2-Wasserstein distance between the decoded distribution and an empirical estimate of the neighbourhood distribution. An extensive experimental analysis is performed, highlighting the benefits of the proposed approach on a range of synthetic datasets to capture structure information. The experimental results also highlight its robustness across 9 different real-world graph datasets (ranging from proximity-oriented to structure-oriented datasets).

Strengths:
- The problem studied is well motivated and the method proposed is well placed in the literature.
- The method is intuitive and the way that the neighbourhood information is reconstructed appears novel.
- The empirical comparisons are extensive.


Weaknesses:
- Some of the choices in matching neighborhoods seem a bit arbitrary and not sufficiently justified.
- The scalability of the proposed method is questionable. The method has a high complexity of O(Nd^3) (where N is the number of nodes and d is the average node degree). The authors address this problem by resorting to the neighborhood sampling method (without citing the prior art), which is only very briefly discussed in the paper.
- The reviewers have also expressed concerns about the fixed sample size q. The question of how the neighbour-sampling is handled when a node has less than q neighbours remains unanswered.